# Identification and Characterization of a Novel Umbra-like Virus, Strawberry Virus A, Infecting Strawberry Plants

**DOI:** 10.3390/plants11050643

**Published:** 2022-02-26

**Authors:** Igor Koloniuk, Jaroslava Přibylová, Radek Čmejla, Lucie Valentová, Jana Fránová

**Affiliations:** 1Department of Plant Virology, Institute of Plant Molecular Biology, Biology Centre, Czech Academy of Sciences, Branisovska 31, 370 05 České Budějovice, Czech Republic; pribyl@umbr.cas.cz (J.P.); jana@umbr.cas.cz (J.F.); 2Research & Breeding Institute of Pomology Holovousy Ltd., Holovousy 129, 508 01 Horice, Czech Republic; radek.cmejla@vsuo.cz (R.Č.); lucie.valentova@vsuo.cz (L.V.)

**Keywords:** novel virus, strawberry, high-throughput sequencing

## Abstract

A novel RNA virus infecting strawberry plants was discovered using high-throughput sequencing. The analyzed plant was simultaneously infected with three different genetic variants of the virus, provisionally named strawberry virus A (StrVA). Although StrVA is phylogenetically clustered with several recently discovered, unclassified plant viruses, it has a smaller genome and several unique features in its genomic organization. A specific and sensitive qPCR system for the detection of identified StrVA genetic variants was designed. A survey conducted in the Czech Republic revealed that StrVA was present in 28.3% of strawberry samples (*n* = 651) from various origins (plantations, gardens, and propagation material). Sequencing of 48 randomly selected StrVA-positive strawberry samples showed that two or all three StrVA genetic variants were present in 62.5% of the samples in various proportions. StrVA was found in mixed infections with other viruses (strawberry mild yellow edge virus, strawberry crinkle virus, strawberry mottle virus, strawberry polerovirus 1, or strawberry virus 1) in 57.1% of the samples, which complicated the estimation of its biological relevance and impact on the health status of the plants.

## 1. Introduction

High-throughput sequencing plays an important role in the discovery of novel viral species that infect plants.

The genus *Umbravirus* encompasses viruses with unipartite positive-sense single-stranded small RNA genomes that encode RNA-dependent RNA polymerase and cell-to-cell and long-distance movement proteins but lack any structural proteins [1,2]. A remarkable ability of umbraviruses is the encapsidation of their genomes inside coat proteins encoded by helper enamoviruses (family *Solemoviridae*). The interactions between these viruses are mutualistic, and enamoviral particles are transported into host tissues by umbraviral movement proteins [3,4]. Several umbravirus–enamovirus pairs have been described thus far [2,3,5]. Notably, experimental systemic infections of umbraviruses may be established in the absence of helper viruses. Umbraviruses are transmitted in a circulative, nonpropagative manner by aphids, which is the result of their encapsidation into the helper’s capsid [2].

Recently, a group of viruses phylogenetically related to umbraviruses was characterized by a wide range of plant host species [6,7,8,9,10,11]. Their genomes displayed several differences in their organization, such as the number of genes, and the encoded proteins did not show similarity to known viral movement proteins. Some of the viruses in this group rely on helper viruses; for example, encapsidation of papaya meleira virus 2 (PMeV2) is dependent on the capsid protein of papaya meleira virus (PMeV), a virus related to mycoviruses from the *Totiviridae* family [7]. Several other molecular and biological properties of the Citrus yellow-vein-associated virus have been studied [9], which, unlike PMeV2, does not rely on any other virus for systemic infection.

Here, we report a novel virus infecting strawberry plants, provisionally named strawberry virus A. In addition to its characterization, an RT–qPCR-based detection system for its diagnosis is provided.

## 2. Results and Discussion

### 2.1. StrVA Genome Sequencing and Characterization

High-throughput sequencing of the 1/2017 sample resulted in approximately 50 million reads (NCBI BioProject accession number PRJNA796705). Upon initial de novo assembly and virome analysis, we found that the 1/2017 plant of *Fragaria vesca* cv. Rujana hosted five different viruses: strawberry crinkle virus (SCV; family *Rhabdoviridae*; genus *Cytorhabdovirus*; two genotypes), strawberry virus 1 (StrV-1; family *Rhabdoviridae*; genus *Cytorhabdovirus*; two genotypes), strawberry polerovirus 1 (SPV1; family *Tombusviridae*, genus *Luteovirus*), strawberry mottle virus (SMoV; family *Secoviridae*; genus *Stramovirus*), and olive latent virus 1 (OLV-1; family *Tombusviridae*; genus *Alphanecrovirus*). The 1/2017 isolates of StrV-1 and SPV1 were described earlier [12,13]. Furthermore, several contigs that resembled umbraviruses were obtained. They shared approximately 90% nucleotide identity with each other, which indicated the presence of several sequentially distinct viral genotypes coinfecting the plant. Variant-specific primers were designed, and complete sequences of three variants of a novel virus, provisionally named strawberry virus A (StrVA), were obtained by Sanger sequencing (GenBank accession numbers MK211273-5). Mapping of HTS reads showed that their mean sequencing coverage ranged from 13 to 22 (Figure 1a).

Three open reading frames were predicted in the StrVA genome (Figure 1b). A slippery motif, GGGAAAC, was located at the end of ORF1. Translation of a fusion protein from partially overlapping ORF1+ORF2 was secured via programmed −1 ribosomal frameshifting and stop-codon readthrough mechanisms similar to those in umbraviruses, as ORF2 lacked a start codon. ORF2 encoded a putative 59 kDa protein that had signatures of RNA-dependent RNA polymerase. Two other putative proteins, p21 and p43, encoded by ORF1 and ORF3, respectively, did not have any orthologues with defined functions in the GenBank database (BLASTP, E-value cut-off: 0.05, search date: 27 December 2021).

The 3′ untranslated region of StrVA was predicted to form two hairpin structures (Figure 2), resembling the H5 and Pr hairpins in carmoviruses, umbraviruses, and the recently described CYVaV. Similarly, the StrVA GGGG motif in the penultimate hairpin was predicted to pair with the 3’ terminal CCCC motif in a ψ1 pseudoknot (analogous to the structure in carmoviruses and tombusviruses [14]). There were some differences in the primary sequence that did not affect the predicted secondary structures (Figure 2b). Interestingly, the 5′ terminus contained a stretch of residues, GGGUAAU, resembling the 5′ GGUAAAU terminus of other umbra- and carmovirus genomic RNAs [15], and was predicted to form a hairpin (Figure 2b).

BLASTN and BLASTX searches of StrVA sequences in a nonredundant database resulted in a list of hits represented by a group of umbravirus-related sequences. Multiple sequence alignment of their replicase sequences showed that StrVA shared moderate amino acid identities (52–57%) with papaya meleira virus 2, babaco virus Q, papaya virus Q, and papaya umbravirus (Figure 3).

It should be noted that the genomic organization of these viruses is characterized by two ORFs divided by ~150 bp-long untranslated regions (Figure 1c). Thus, they clearly lack a predisposition for translational ribosomal frameshifting, unlike StrVA. Furthermore, their genomes have much longer 3′ UTRs than the shorter ones found in StrVA. Interestingly, phylogenetic analysis of the polymerase protein set these viruses and StrVA apart from another group consisting of the recently described Citrus yellow-vein associated virus [9] and other viruses (Figure 4).

Notably, two main differences were observed: ribosomal frameshifting was predicted only for Citrus yellow-vein-associated virus, opuntia umbra-like virus, and sugarcane umbra-like virus, and the putative slippery sequence, GGGUUUU, was different from that of StrVA (GGGAAAC). These and the abovementioned differences highlight the heterogeneity of this group of viruses and show that, despite phylogenetic relationships and moderate sequence similarities, they likely belong to different evolutionary lineages.

### 2.2. Real-Time PCR for StrVA Detection

For detection purposes, a quantitative PCR assay was developed. The specificity of the StrVA real-time PCR design was tested by using two pairs of primers that were in a nested configuration. The outer primers were used for endpoint PCR and sequencing verification of amplified PCR products; the inner primers were combined with a probe and used for real-time PCR detection of StrVA. The same results were obtained for both primer sets (data not shown). The sequencing results for 48 samples validated all positive findings.

Using a synthetic standard, a sensitivity of 50 copies per reaction was reached with confidence (Figure 5a), while 5 copies per reaction could be amplified in 25% of the tested diluates. The characteristics of the standard curve are shown in Figure 5b.

### 2.3. StrVA Presence in Strawberry Samples

In total, 651 strawberry samples (622 samples of *Fragaria* × *ananassa* Duchesne and 29 samples of *F. vesca*) were analyzed for the presence of StrVA and other viruses, such as strawberry mild yellow edge virus (SMYEV), SCV, strawberry vein banding virus (SVBV), SMoV, SPV1, and StrV-1. Since StrVA represents a novel virus, strawberry propagation material from various origins was also included in the survey, as well as samples from commercial plantings (27 plantations in 6 regions) and gardens (5 gardens) (Table 1). 

Most StrVA-positive samples were found in commercial plantings and gardens (62.8% and 21.4%, respectively), but StrVA was also detected in certified propagation material. One of the reasons for StrVA negativity in hobby markets’ materials might be due to rather small batches of plants—there were a total of 35 varieties originating from different suppliers. On the other hand, CAC2-certified propagation materials were obtained from a single supplier. Since all the plants were asymptomatic and StrVA was unknown until now, the virus might spread in the production facility. The presence of other known strawberry viruses is regularly checked by the state extension service and is thus kept under control, likely explaining the fact that no other viruses were present.

Overall, StrVA was detected in 28.3% of all samples, with the highest loads being approximately 100 million StrVA copies per mg of fresh leaf tissue. StrVA was found in coinfections with other viruses in 39.1% of samples and was most often found in mixed infections with StrV-1 (25%) and SPV1 (23.6%).

### 2.4. Single-Host Infection with Multiple Genetic Variants of StrVA

The 1/2017 plant of *F. vesca* cv. Rujana hosted three divergent variants of StrVA. Their whole-genome nucleotide identities ranged from 76% to 91%, with variants A and C being more similar (91%). Interestingly, the differences were interspersed across all genomic positions (Figure 6).

Most nucleotide differences were of synonymous nature, which is demonstrated by the higher values of identities of ORF1+ORF2 and ORF3 products (Table 2, Figure 6b). An exception is the lower similarity of the p43 protein of StrVA, var. B, where only 70 % amino acid identity was conserved with p43 proteins of either A or C variants. Nearly all differences in p43 between the StrVA variants were located in the C-terminal part of the protein (Figure 6b).

Such high divergence of genomic sequences is not unique among viruses infecting strawberry plants. For example, three variants of a rhabdovirus StrV-1 coinfecting the same plant shared 76–87% of nt identity in their genomes [12]. At the same time, the strawberry plant was hosting three RNA1 and three RNA2 variants of a stramovirus, SMoV (GenBank accession numbers MH013322-7). The nt conservation between the variants was 79–81% (RNA1) and 77–85% (RNA2). Similar to StrVA, a vast majority of nt changes between those viral variants were synonymous.

During the validation of genome sequences, we performed mappings with less stringent settings (minimum 50% of read length and 97% sequence identity) and did not visually detect breakpoints in the resulting mappings. Subsequent analyses with the recombination detection program RDP5 did not reveal any recombination breakpoints between the three sequenced genomes that were significantly supported by more than five algorithms implemented in the program (Appendix A).

Further evidence that such coinfections with several StrVA variants are not rare was gathered during the validation of the RT–qPCR results with Sanger sequencing (Figure 6, red annotation). Based on the presence of double peaks, 31 out of the 48 samples contained more than one StrVA genotype. However, 17 sequences did not contain double peaks (representative isolates shown in Figure 7; for all isolates, see Appendix A). Only one isolate was assigned to the B genotype, whereas the rest were equally divided between the A and C groups.

### 2.5. Detection of StrVA in Non-Strawberry Hosts

In total, 42 non-*Fragaria* plants growing in close proximity to strawberry plants were tested using the RT–PCR StrVA assay. Only one plant of marsh yellow cress, *Rorippa palustris*, was identified as a natural StrVA host. The plant tested negative for SPV1, SMoV, OLV1, SCV, StrV1, SMYEV, and SVBV. Marsh yellow cress, a plant of the family *Brassicaceae*, is found on river banks and is a common weed in wet arable fields [16]. Sequence analysis of the StrVA isolate from the marsh yellow cress and the isolates from nearby strawberries showed 100% nucleotide identity, suggesting that either of them might serve as a natural StrVA reservoir.

## 3. Conclusions

In this study, we report a novel virus, StrVA, infecting strawberry plants. Based on phylogenetic and sequence evidence, StrVA is related to a group of yet-to-be-classified viruses that display diverse characteristics, including the number of genes, the length of their genomes, and regulatory elements (−1 ribosomal frameshifting). In addition to strawberry plants, another natural host, marsh yellow cress (*Brassicaceae*), was identified during the StrVA survey, indicating that the virus’s host range may not be narrow. Over 600 strawberry samples were screened, and StrVA was found in plants from diverse sources, including certified plant material, thus confirming that the virus has spread widely, at least in the Czech Republic. More than one-third of StrVA-positive plants were coinfected with other viruses. Further effort should be directed to examining StrVA vectors, identifying any putative helper viruses, and determining their biological significance.

## 4. Materials and Methods

### 4.1. Plant Materials

An *F. vesca semperflorens* cv. Rujana 1/2017 plant was cultivated from commercially available seeds at a private garden in South Bohemia (locality Třísov, Czech Republic) in spring 2016, and a leaf sample was taken one year later. For virus screening, strawberry samples were collected throughout the Czech Republic and came from various sources: commercial plantings, gardens, SE-certified propagation material, CAC-certified propagation material, and plants from hobby markets.

Other plants tested for the presence of StrVA included *Aegopodium podagraria* L. (n = 1), *Aster* sp. (n = 1), *Arabidopsis thaliana* (L). Heynh. (n = 1), a wild seedling of *Betula pendula* Roth (n = 1), *Carex* sp. (n = 1), *Chenopodium* sp. (n = 1), *Epilobium parviflorum* Schreb. (n = 1), *Hypochaeris radicata* L. (n = 1), *Malus* sp. (n = 10), *Poa annua* L. (n = 1), *Potentilla* sp. (n = 1), *Prunus* sp. (cherry: n = 2, plum: n = 1), *Rorippa palustris* (L.) Besser (n = 1), *Rumex obtusifolius* L. (n = 1), a wild seedling of *Salix caprea* L. (n = 1), *Scorzoneroides autumnalis* (L.) Moench. (n = 1), *Sonchus arvensis* L. (n = 4), *Stellaria media* (L.) Vill. (n = 5), and *Taraxacum officinale* Web. (n = 6).

### 4.2. Sampling, Real-Time Reverse Transcription PCR and Reverse Transcription PCR Detection of Viruses

Total RNA was isolated from 50 mg of tested plant leaves using a Ribospin Plant Kit (GeneAll Biotechnology, Seoul, Korea) according to the manufacturer’s instructions. M-MLV Reverse Transcriptase (Invitrogen, Waltham, MA, USA) was used for reverse transcription, and 1 µg of RNA was added as a template.

Genomic termini of each StrVA variant were independently sequenced using 5′- and 3′-RACE kits (Invitrogen, Waltham, MA, USA) with variant-specific primers (Appendix A). For the determination of 3′ termini, the total RNA was previously polyadenylated with poly(U) polymerase and adenosine triphosphate (NEB, Ipswich, MA, USA) following the manufacturer’s recommendations. The obtained products were purified and directly sequenced (Eurofins Genomics, Luxembourg).

For StrVA PCR detection, sequences obtained from NGS were used to design two pairs of primers and a probe in a putative RdRp region (Table 3, see Appendix A for the complete list of primers used in this study). One pair of primers was selected to analyze the sequence diversity of this region, and the second pair and a HEX-labelled probe were used for real-time PCR for StrVA detection (Table 3). To assess the detection sensitivity, a synthetic Ultramer™ DNA Oligonucleotide (Integrated DNA Technologies Coralville, IA, USA) was used in a serial dilution. PCR conditions were the same as those described below.

Plants were also tested for the presence of the following viruses by real-time PCR: StrV-1, SMYEV, SCV, SVBV, SMoV, and SPV1. Expression of the mitochondrial Nad5 gene was used as an internal control. The following PCR conditions were used for all viruses: newly designed forward primers, reverse primers, and probes labelled with FAM at final concentrations shown in Table 3; 10 µL of qPCR 2× Blue Master Mix (Top-Bio, Vestec, Czech Republic); 2 µL of cDNA as a template; and water to a final volume of 20 µL. Real-time PCR was run on a Rotor-Gene Q cycler (Qiagen, Hilden, Germany) under the following conditions: initial denaturation at 94 °C for 5 min; 50 cycles of denaturation at 94 °C for 20 s denaturation; annealing at 58 °C for 20 s; and elongation at 72 °C for 20 s.

Reverse transcription PCR for the determination of non-strawberry hosts was performed using the above-described cDNA preparation step. Then, 1 µL of cDNA was mixed with 10 µL of 2× PPP Master Mix (Top-Bio), 8 µL of PCR-grade H_2_O, and the 1750:1751 primers (Table 3). The primers used in the study were those previously published for SMoV, SCV, SMYEV, SVBV [17], SPV1 [18] and StrV-1 [12]. Reaction mixtures devoid of cDNA templates served as no-template controls. Each PCR product (4 µL) was analyzed by electrophoresis in a 1% agarose gel prestained with GelRed (Biotium, Hayward, CA, USA).

Sanger sequencing was performed using purified PCR products mixed with the appropriate primers (Eurofins Genomics, Luxembourg).

The sequencing library from total RNA with a preceding RiboZERO (Illumina, San Diego, CA, USA) treatment was prepared using the NEBNext Ultra II Directional RNA Library Prep Kit for Illumina (NEB) and then processed on a HiSeq 4000 platform in 100 b SE output mode (SEQme s.r.o., Dobris, Czech Republic).

### 4.3. Sequence Analyses

All analyses were performed using Geneious Prime^®^ 2022.0.1 (Biomatters, Auckland, New Zealand) and CLC Genomics Workbench 9.5.1 (Qiagen). High-throughput data were trimmed and then de novo assembled using CLC Genomics Workbench with a minimum contig length of 200 nt. All obtained contigs were analyzed using the BLASTX tool in Geneious software against a custom local database of viral proteins. Furthermore, virus-like sequences were analyzed against the GenBank database (11 November 2021) with BLASTX (cut-off E value 0.005). Sanger sequences were processed using Geneious software.

Nucleotide sequences and in silico translated sequences were compared using BLAST+ [33] against GenBank (November 2021) and custom local databases. The analysis was performed on the Phylogeny.fr platform and comprised the following steps. Sequences were aligned with MUSCLE (v3.8.31) configured for the highest accuracy (MUSCLE with default settings). After alignment, positions with gaps were removed from the alignment. The phylogenetic tree was reconstructed using the maximum likelihood method implemented in the PhyML program (v3.1/3.0 aLRT). The reliability of the internal branch was assessed using the aLRT test (minimum SH-like and Chi2-based parametric). Putative recombination events were detected and evaluated in a program RDP5 [19] using the MUSCLE multiple alignment of the complete StrVA genome sequences.

## Figures and Tables

**Figure 1 plants-11-00643-f001:**
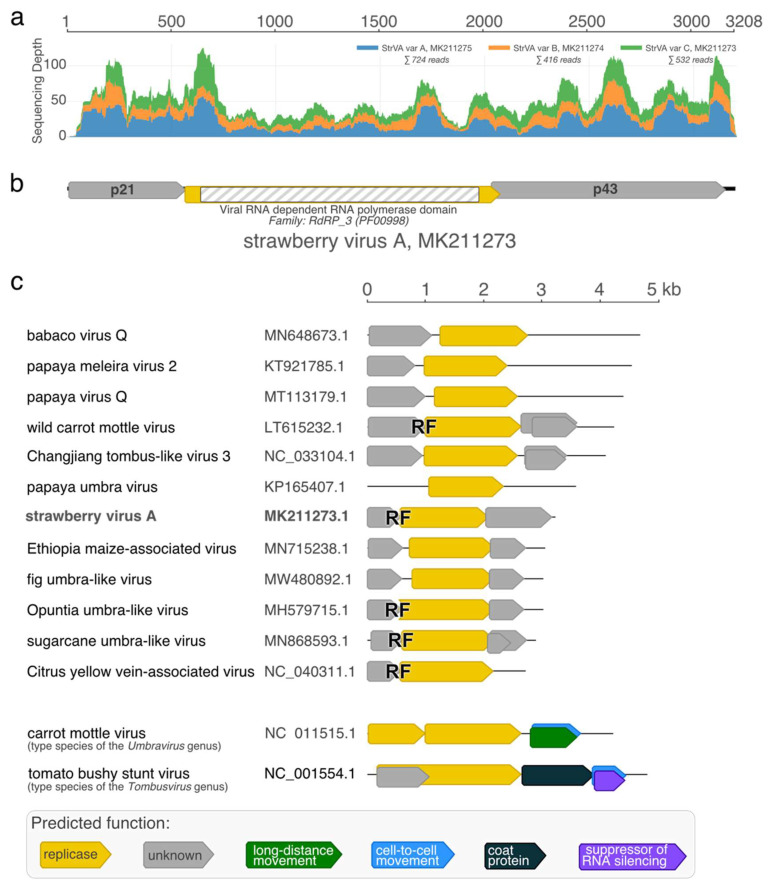
Sequencing coverage (**a**), genomic organization of strawberry virus A (StrVA) (**b**), and StrVA-related viruses (**c**). RF stands for predicted −1 ribosomal frameshifting.

**Figure 2 plants-11-00643-f002:**
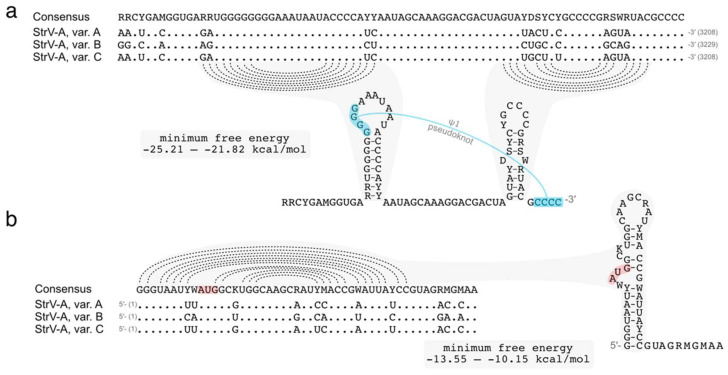
Putative hairpin structures at the termini of the StrVA genome. Multiple alignment and secondary structure of the consensus sequences of the 3′ (**a**) and 5′ (**b**) ends. Dotted lines show predicted pairing. The putative 3′ ψ1 pseudoknot is annotated in blue.

**Figure 3 plants-11-00643-f003:**
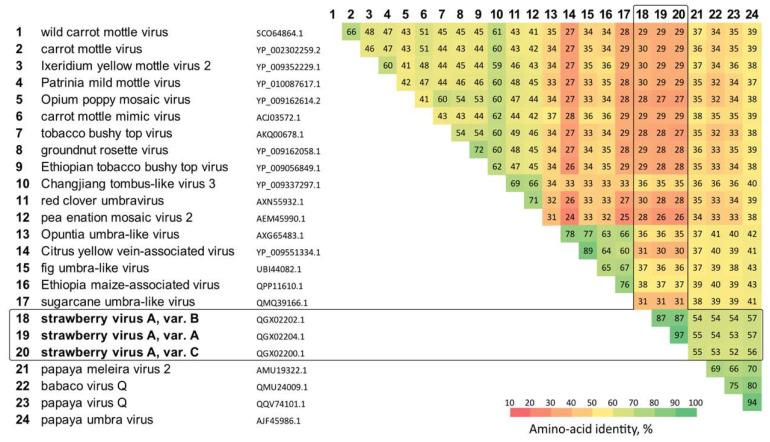
Conservation of amino acid identities in the polymerase gene of StrVA and related viruses.

**Figure 4 plants-11-00643-f004:**
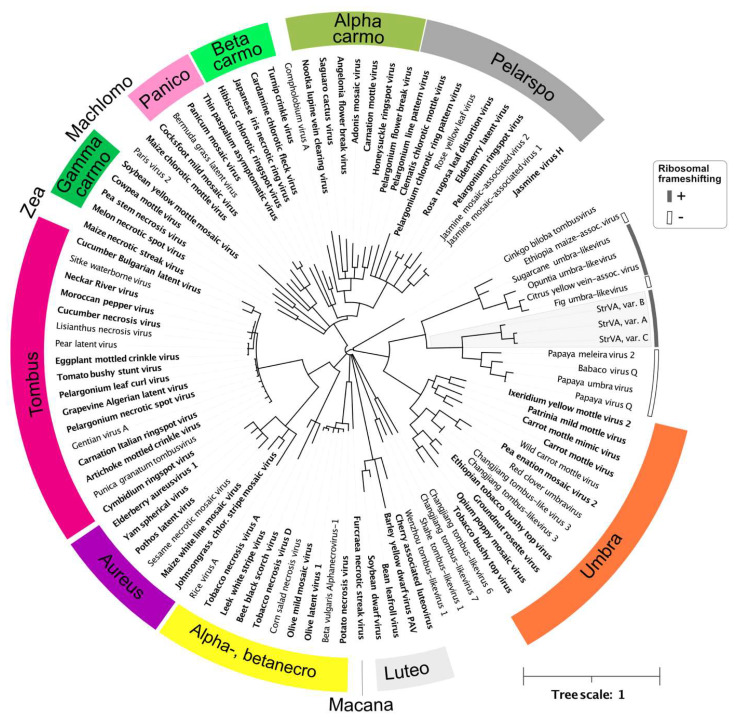
Phylogeny of StrVA, tombusviruses, and other related viruses based on the RNA polymerase protein. All branches with less than 0.9 support are collapsed. The scale is shown in the lower right corner. Viruses officially recognized as species by ICTV are in bold; genera are color-coded. The accession numbers of the sequences used are shown in Appendix A. For StrVA and related viruses, putative ribosomal frameshifting is annotated as shaded or unshaded arcs.

**Figure 5 plants-11-00643-f005:**
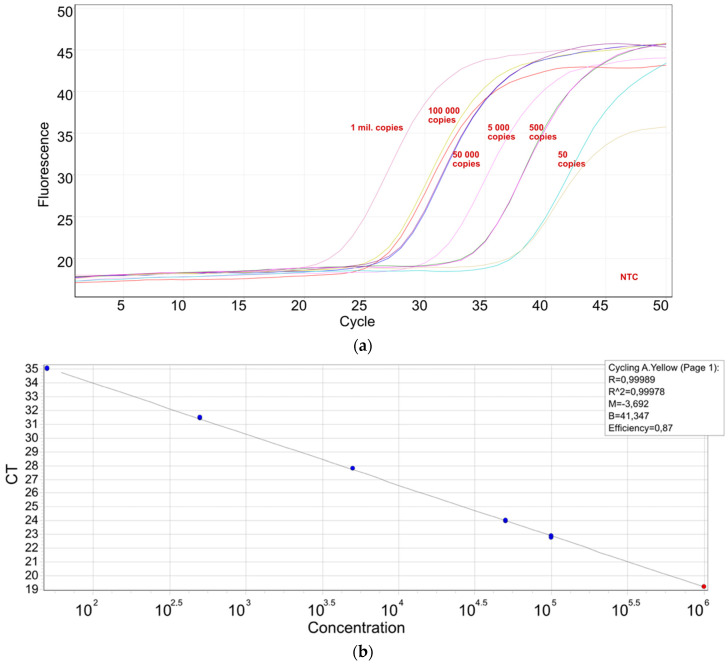
A real-time PCR system for StrVA detection: (**a**) a synthetic standard was used to assess the sensitivity of the real-time PCR used for the detection of StrVA; (**b**) standard curve parameters.

**Figure 6 plants-11-00643-f006:**
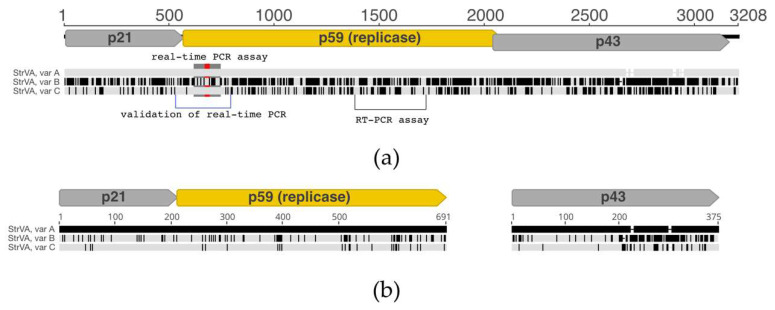
Multiple nucleotide sequence alignment of the three variants of StrVA infecting the 1/2017 strawberry plant. StrVA, var. A was selected as the reference; all differences from the reference are in black. (**a**) Alignment of nucleotide sequences of complete genomes. Regions used for the estimation of StrVA variability and validation of qPCR and RT-PCR assays are annotated. (**b**) Alignments of amino acid sequences of p21+p59 and p43 proteins.

**Figure 7 plants-11-00643-f007:**
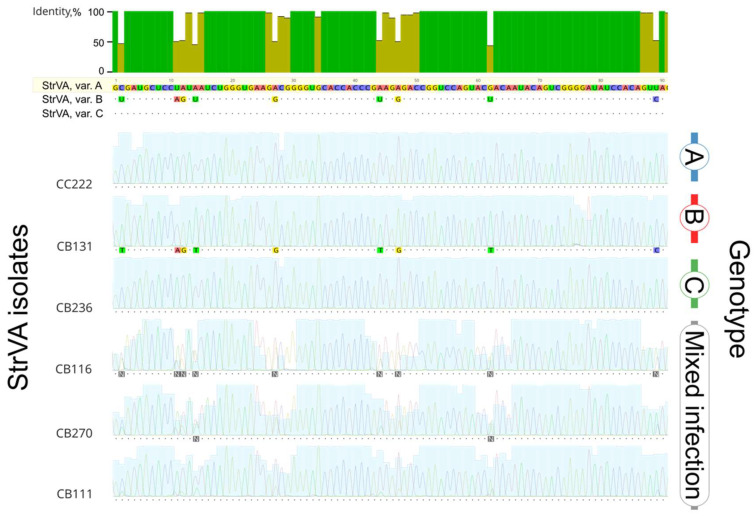
Multiple sequence alignment of the replicase gene region (genome positions 610–699) of several StrVA isolates (all 48 isolates are shown in Appendix A). StrVA variant A was chosen as the reference. For other sequences, matching bases are shown as dots, and differences are highlighted. Multiple peaks indicate the presence of more than one StrVA genotype.

**Table 1 plants-11-00643-t001:** StrVA findings in strawberry samples.

Category	Samples (n, %)	StrVA Positivity (n, %)	Mixed Infections (n, %)
Commercial plantings	234 (35.9%)	147 (62.8%)	62 (42.2%)
Gardens	42 (6.5%)	9 (21.4%)	8 (88.9%)
SE ^1^-certified propagation material	80 (12.3%)	3 (3.8%)	2 (66.7%)
CAC ^2^-certified propagation material	186 (28.6%)	25 (13.4%)	0
Hobby markets	109 (16.7%)	0	0
Total	651	184 (28.3%)	72 (39.1%)

^1^ SE—state extension service certification; ^2^ CAC—‘Conformitas Agraria Communitatis’ propagation material.

**Table 2 plants-11-00643-t002:** Percentages of conserved nucleotide and amino acid (bold in parentheses) identities of StrVA’s ORFs and putative protein between viral variants coinfecting the 1/2017 strawberry plant.

**Virus**	StrVA, var. A	StrVA, var. B	StrVA, var. C	
StrVA, var. A	-	69 (**70**)	89 (**91**)	ORF3 (p43)
StrVA, var. B	80 (**87**)	-	71 (**70**)	
StrVA, var. C	92 (**97**)	92 (**87**)	-	
	ORF1+ORF2 (p21 and p59.4)			-

**Table 3 plants-11-00643-t003:** List of primers used in the study.

Primer/Probe	Sequence (5′ to 3′ Direction)	Final Concentration (μM)	Usage
StrVA			
Forward	ACCTGGCCTTGTCYCGGC	0.5	sequencing of virus isolates
Reverse	WGGTGGWGGGGACGTACAAC	0.5	
1750 Forward	GATGTCTGGTGATATGGACA	0.5	RT-PCR assay
1751 Reverse	ACCAATTCTCTACATCGTGT	0.5	
Forward	ATCTGGGTGAAGRCGGGGT	0.5	qPCR
Reverse	CAAAACCCTCTCATAAGGTTRTCCA	0.5	
Probe	ACTGTGGATATCCCCGACTGTATTGT	0.2	
StrV-1			
Forward	AACGGATATTGTGGCGCRAA	0.5	qPCR
Reverse	CCTGATGTTGTTKATATARCTGAGRTC	0.5	
Probe 01	AAACCTCTTACCATCATCTCGTAAA	0.175	
Probe 02	AARCCCCTCACCATCATYTCGTAA	0.175	
SMYEV			
Forward	CCCTCCTGACGTACACAACAACTG	0.5	qPCR
Reverse 01	CCGTGAGGGAGGAGAATACGC	0.25	
Reverse 02	CCGTGAGGGAGGAGAATACAC	0.25	
Probe	TACTCTAGTYGCCATCGAGGTACAGTGC	0.2	
SCV			
Forward 01	ACAGTRTGCGCTTTAGAGGTTGTT	0.5	qPCR
Forward 02	ACAGTGTGCGCTTTAGAGGTTATTC	0.5	
Forward 03	ACAGTGTGCGCTTTAGAAGTTGTTC	0.25	
Reverse 01	ACCTGATTATCTCCCATYCCCATT	0.5	
Reverse 02	ACTTGATTATCCCCCATCCCCA	0.5	
Probe 01	TCTCAATAYGATTGTACATACCGCAT	0.15	
Probe 02	TCTCAATACGATTGCACATATCGCAT	0.15	
SVBV			
Forward	AATATCTGTCTTTACTTGATSATGAACTTG	0.5	qPCR
Reverse 01	CGTCTTCGCTGCTGCTGTAGTC	0.25	
Reverse 02	CATCTTCACTGCTGCTGCTGTAG	0.25	
Probe	AGTTACAGGTACTTGTAGCAAAAGARATGA	0.2	
SMoV			
Forward	GTAGGACACCGGCTCTTGGYAGT	0.5	qPCR
Reverse	TTGGRTCGTCACCTGAYCTCG	0.5	
Probe	ACAGGWGGCACTGTTTACAGTGTTCC	0.25	
SPV1			
Forward	CAACTGGGGTCGTACACTCGC	0.5	qPCR
Reverse	GGCCAGCCGAATCCTTTGAC	0.5	
Probe	ACCTACCCTGAACTCGCCGAACA	0.2	
Nad5			
Forward	GATGCTTCTTGGGGCTTCTTGTT	0.25	qPCR
Reverse	ACATAAATCGAGGGCTATGCGG	0.25	
Probe	CCACAATTAACATCACTACGGTCGGGCTA	0.2	

## Data Availability

Sequence data were deposited under NCBI BioProject accession number PRJNA796705 and nucleotide sequence records MK211273-5.

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
