# Peer review of "Identification and Characterization of a Novel Umbra-like Virus, Strawberry Virus A, Infecting Strawberry Plants"

_plants, 2022, doi:10.3390/plants11050643_

Round 1

Reviewer 1 Report

This is a solid paper describing identification of StrVA, a novel umbra-like virus in strawberry, and showing its wide occurrence in fields. Additionally, the possibility of StrVA transmission to a test plant has been demonstrated, and StrVA was found in a common weed Rorippa palustris, which can be a natural virus reservoir. To my mind, the paper is worth publishing. However, I have some concerns, which should be addressed before the paper can be accepted for publication

Major concerns

  1. The three variants of StrVA should be discussed in more detail. It should be explained what species demarcation criteria were used when the three sequences were decided to be considered strains of one virus rather than three distinct viruses. In particular, if the nucleotide identity between a pair of these strains is only 76% (line 192), should these strains be considered as different viruses? Current ICTV guidelines for virus species demarcation should be followed. Besides, it would be interesting for the readers to know whether recombinants between the three ‘strains’ could be detected.
  1. Fig. 4. should be modified to make its central part (the tree) more readable. I would suggest using much smaller font size for the genera names and removing the thick color arcs denoting genera. Instead, sectors corresponding to genera can be colored. This would give a space for enlarged tree.
  1. As the authors prefer the combined ‘Results and Discussion’ format, and there is no true Discussion in the paper, I think that Conclusions, which put the reported data in the current context, should be added after ‘Results and Discussion’.

Minor concerns

Lines 30-32. Many recent...  - The phrase is not clear, should be re-written or removed.

Line 38. enamoviral - has not been defined so far.

Line 41.  the helper virus - should be ‘helper viruses’.

Lines 43-44. The phrase should be re-written

Line 50.  - properties of what?

Line 56. It is not clear what “the sample” is.

Lines 66-67. several sequentially distinct viral genotypes - can be: several related, clearly distinct viruses… Note that ‘genotypes infect’ is incorrect.

Lines 90-91. GGGUAAU, that is partially conserved with the 5’ GGUAAAU terminus of other umbra- and carmoviruses-

can be: GGGUAAU resembling the 5’ GGUAAAU terminus of other umbra- and carmovirus genomic RNAs

Line 191. sequentially divergent - should be corrected

Author Response

Dear reviewer, thank you for reading our manuscript and your comments. Please, find below a point-by-point response to the raised questions.   Major concerns 1The three variants of StrVA should be discussed in more detail. It should be explained what species demarcation criteria were used when the three sequences were decided to be considered strains of one virus rather than three distinct viruses. In particular, if the nucleotide identity between a pair of these strains is only 76% (line 192), should these strains be considered as different viruses? Current ICTV guidelines for virus species demarcation should be followed. Besides, it would be interesting for the readers to know whether recombinants between the three ‘strains’ could be detected.   ## Response ##  One of problems with demarcation criteria is that they exist neither for StrVA nor for related viruses and we can adopt them from the most related genus - Umbravirus. We include more detailed description and comparison of available three StrVA sequences. Briefly, the majority of nucleotide changes are synonymous and do not lead to changes of the encoded proteins. Recombination analysis was included to the text, its results were appended as Table S3. Two detected recombination events had weak support (detected by either three or two algorithms out of total eight algorithms implemented in RDP). Further, during validation of genome sequences we have performed mappings with less stringent settings (minimum 50% of read should have 97% identity) and have not detected breakpoints in the resulting mappings. However, it should not be excluded that during coinfection interactions between sequence variants may occur. Additionally, at this point there is certain lack of full length StrVA genomes to conduct comprehensive recombination study.   2Fig. 4. should be modified to make its central part (the tree) more readable. I would suggest using much smaller font size for the genera names and removing the thick color arcs denoting genera. Instead, sectors corresponding to genera can be colored. This would give a space for enlarged tree.   ## Response ##  The central part of the tree was scaled up and modifications of the genera names were made.   3As the authors prefer the combined ‘Results and Discussion’ format, and there is no true Discussion in the paper, I think that Conclusions, which put the reported data in the current context, should be added after ‘Results and Discussion’.   ## Response ##  the conclusion part was moved at the end of the Results & Discussion.   Minor concerns Lines 30-32. Many recent...  - The phrase is not clear, should be re-written or removed. ## Response ##  the sentence was removed as there is a mention of such group of viruses below.   Line 38. enamoviral - has not been defined so far. ## Response ##  'poleroviruses' mentioned above should be 'enamoviruses'; a correction was made   Line 41.  the helper virus - should be ‘helper viruses’. ## Response ##   corrected   Lines 43-44. The phrase should be re-written ## Response ##  the sentence was modified   Line 50.  - properties of what? ## Response ##  corrected   Line 56. It is not clear what “the sample” is. ## Response ##  corrected     Lines 66-67. several sequentially distinct viral genotypes - can be: several related, clearly distinct viruses… Note that ‘genotypes infect’ is incorrect. ## Response ##  this part was reformulated to acknowledge that viruses belonged to five species.   Lines 90-91. GGGUAAU, that is partially conserved with the 5’ GGUAAAU terminus of other umbra- and carmoviruses- can be: GGGUAAU resembling the 5’ GGUAAAU terminus of other umbra- and carmovirus genomic RNAs ## Response ##  the suggestion was accepted    Line 191. sequentially divergent - should be corrected ## Response ##  'sequentially' was removed

Reviewer 2 Report

The article of Koloniuk et al. entitled „Identification and characterization of a novel umbra-like virus, strawberry virus A, infecting strawberry” is the first report of a new virus identified in strawberry plants by high-throughput sequencing (HTS) and identified also by RT/PCR and nucleotide sequence determination. Three different isolates of StrVA was identified according to nt sequence and phylogenetic analysis. An RT/PCR method for the StrVA detection was also developed.

Generally the identification and characterization of new viruses is important topic regarding to the general knowledge and also in economic impact. Umbra-like viruses are not among the viruses characterized in detail, so in this respect all the new data are interesting.

The original strawberry plant used for HTS designated as “1/2017 isolate of Fragaria vesca cv. Rujana”. Since in virological studies the “isolate” usually refer virus strains, I suggest to use plant instead of isolate through out of the study to avoid misunderstanding.

The transmission experiments are generally confusing. The symptoms on Nicotiana occidentalis test plants are demonstrated only seven days after the inoculation (Fig 5), but not 14 days after the transfer when systemic symptoms could be observed (ln135). If this experiment is included in the MS, please include the photos 14 days after the infection. Unfortunately StrVA was not identified in aphids (ln 16-148) and even two-spotted spider mite (Tetranychus urticae) sequences were identified in the HTS data. So the results are not convincing in this respect, the vector of the transmission is not clearly proved, so I suggest to delete this part of the MS.

Please verify the PCR detection method in the case of the different StrVA strains (var A, B and C) (ln 159-170).

The presence of StrVA was verified in total 651 strawberry samples. It is really interesting, that plants originated from hobby markets were virus free. What could be the reason? In the case of CAC2-certified propagation materials just the StrVA was present and no other virus. What could be the reason of this observation? Were there any visible sign of virus infection on these plant? The pathological significance of the virus would be interesting, and these plants could enhance the analysis in this respect.

In non-strawberry host (ln 211-217) just StrVA was tested. It would be interesting to analyse these plants for the presence of the other viruses tested previously in strawberry samples (Table 1).

Generally the MS is interesting and I suggest to publish it in Plants after the proposed modifications.

Minor points:

ln 33: Pleasi insert: “positive-sense single-stranded…”

ln 46: Change signatures to similarity

ln 69: identified instead of obtained

ln 80: Please detail the features of 59. kDa 

Author Response

Dear Reviewer, thank you for reading the manuscript and pointing out mistakes. Please find below our point-to-point response. During writing we saw it as the most confusing  section of the manuscript. At that time, we have decided to keep it as there is/was no available data on potential vectors of StrVA-like viruses. So, the major change in the current revision was complete removal of the part describing the virus transmission.   The original strawberry plant used for HTS designated as “1/2017 isolate of Fragaria vesca cv. Rujana”. Since in virological studies the “isolate” usually refer virus strains, I suggest to use plant instead of isolate through out of the study to avoid misunderstanding.   ## Response ##  the suggested changes were made throughout the whole manuscript   The transmission experiments are generally confusing. The symptoms on Nicotiana occidentalis test plants are demonstrated only seven days after the inoculation (Fig 5), but not 14 days after the transfer when systemic symptoms could be observed (ln135). If this experiment is included in the MS, please include the photos 14 days after the infection. Unfortunately StrVA was not identified in aphids (ln 16-148) and even two-spotted spider mite (Tetranychus urticae) sequences were identified in the HTS data. So the results are not convincing in this respect, the vector of the transmission is not clearly proved, so I suggest to delete this part of the MS.   ## Response ##  The transmission part was removed.   Please verify the PCR detection method in the case of the different StrVA strains (var A, B and C) (ln 159-170).   ## Response ##  The qPCR primers and the probe were designed based on the consensus sequence from multiple alignment of 48 StrVA isolates.      The presence of StrVA was verified in total 651 strawberry samples. It is really interesting, that plants originated from hobby markets were virus free. What could be the reason? In the case of CAC2-certified propagation materials just the StrVA was present and no other virus. What could be the reason of this observation? Were there any visible sign of virus infection on these plant? The pathological significance of the virus would be interesting, and these plants could enhance the analysis in this respect. ## Response ##  Over 100 plants were bought in 14 various hobby markets; plants were of 35 varieties, and were sold bare rooted, as frigo plants, or in containers; plantlets were supplied by various producers.  This may be one of many reasons that no StrVA was identified in this type of plant material – only small batches of plants of various origins were tested. Another reason may be that propagation material producers follow good manufacturing practice that minimizes the chance plantlets can get infected. On the contrary, CAC2-certified propagation materials originated from only one supplier. Since all plants were asymptomatic and StrVA was unknown until now, the virus might spread in the production facility. The presence of other known strawberry viruses are regularly checked by a control authority, and are thus kept under control, likely explaining the fact no other viruses were present. The pathological/economic significance of StrVA remains to be assessed in future works. We have added this information to the text.     In non-strawberry host (ln 211-217) just StrVA was tested. It would be interesting to analyse these plants for the presence of the other viruses tested previously in strawberry samples (Table 1).   ## Response ##  The StrVA-positive plant was tested negative for .... SPV1, SMoV, OLV1, SMYEV, SCV, StrV1, and SVBV. The information was added to the manuscript.   Generally the MS is interesting and I suggest to publish it in Plants after the proposed modifications.       Minor points:   ln 33: Pleasi insert: “positive-sense single-stranded…” ## Response ##     ln 46: Change signatures to similarity ## Response ##   corrected   ln 69: identified instead of obtained ## Response ##   corrected   ln 80: Please detail the features of 59. kDa  ## Response ##  details of domain search were added; we have considered that description of four RNA polymerase motifs would be unnecessary detailed here